# ADAPTIVE Q-LEARNING FOR INTERACTION-LIMITED REINFORCEMENT LEARNING

## ABSTRACT

Conventional reinforcement learning (RL) needs an environment to collect fresh data, which is impractical when an online interaction is costly. Offline RL provides an alternative solution by directly learning from the logged dataset. However, it usually yields unsatisfactory performance due to a pessimistic update scheme or/and the low quality of logged datasets. Moreover, how to evaluate the policy under the offline setting is also a challenging problem. In this paper, we propose a unified framework called Adaptive Q-learning for effectively taking advantage of offline and online learning. Specifically, we explicitly consider the difference between the online and offline data and apply an adaptive update scheme accordingly, i.e., a pessimistic update strategy for the offline dataset and a greedy or no pessimistic update scheme for the online dataset. When combining both, we can apply very limited online exploration steps to achieve expert performance even when the offline dataset is poor, e.g., random dataset. Such a framework provides a unified way to mix the offline and online RL and gain the best of both worlds. To understand our framework better, we then provide an initialization following our framework. Extensive experiments are done to verify the effectiveness of our proposed method.

## 1 INTRODUCTION

Conventional online reinforcement learning (RL) methods (Haarnoja et al., 2018; Fujimoto et al., 2018) usually learn from experiences generated by interactions with the online environment. They are impractical in some real-world applications, e.g., dialog (Jaques et al., 2019) and education (Mandel et al., 2014), where interactions are costly. Recently, offline RL (Levine et al., 2020) has aroused much attention. It targets the above challenge by making the agent learn from an offline dataset collected by other policies in a purely data-driven manner. The difference between online RL and offline RL is shown in Figure 1.

Existing offline RL studies try to target the distribution mismatch or out-of-distribution actions issue by employing a pessimistic update scheme (Kumar et al., 2019; 2020) or in combination with imitation learning (Fujimoto et al., 2019). However, when the dataset is fixed, offline RL cannot learn the optimal policy (Kidambi et al., 2020), and even worse, when the dataset's quality is poor, offline RL usually gains a relatively bad performance (Kumar et al., 2020; Fu et al., 2020; Levine et al., 2020). On the other hand, it is challenging to evaluate the learned policy when learning totally from the offline dataset. Even though some research topics, e.g., off-policy evaluation (Dann et al., 2014), study how to evaluate the learned policy without the interaction with the environment, it is still not ideal for the practical purpose.

Some recent works try to address the above issues by employing an offline-online setting. Such methods (Lee et al., 2021; Nair et al., 2020) focus on pre-training a policy using the offline dataset and fine-tuning the policy through further online interactions. Even though their methods alleviate the above issues to some extent, their main bottleneck is that they do not consider the different characteristics of offline and online data. For instance, pre-existing offline data can prevent agents from converging prematurely due to the potential diverse offline dataset, while online data can improve stability and accelerate convergence. Generally, these different data are mixed and used by a pessimistic strategy to update the policy in their methods, which may be problematic since using a pessimistic strategy for online data may harm the policy performance (Nair et al., 2020). Moreover,

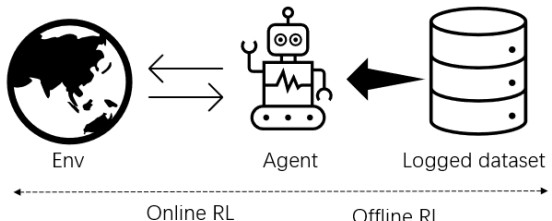

Figure 1: Online RL collect the data by interacting with the environment and they don't utilize the existing logged dataset while offline RL only exploit logged dataset without any future performance improvement. By contrast, we focus on obtaining the best of both worlds.

with sufficiently large and diverse offline data, a high-performing policy can be learned just using a pure online RL algorithm (Agarwal et al., 2020). And the online near-on-policy data also play a key role in improving the RL algorithm's stability (Fujimoto et al., 2019). Hence, we should take full advantage of both offline and online dataset.

To tackle the above problems, in this paper, we emphasize that *online and offline RL should be coupled organically*. First, a separate updating strategy should be employed for online and offline data, respectively, considering their different characteristics. To do so, we present a framework called adaptive Q-learning that integrates the advantage of offline learning and online learning effectively. When learning from the offline dataset, we conduct a pessimistic update strategy. In contrast, we use a greedy or non-pessimistic update strategy when learning from the online dataset. Second, we design a novel replay buffer to distinguish the offline from online datasets in a simple way. By utilizing such a novel framework and buffer design, the agent can achieve an expert policy using limited online interaction steps regardless of the quality of the offline dataset. In the experiments, our proposed framework can achieve better performance by using only one fifth number of interactions compared with the previous method (Nair et al., 2020).

Our contributions can be summarized as below:

- We propose a unified framework called Adaptive Q-learning that can effectively benefit from both the offline dataset and limited number of online interaction data.
- Based on the general framework, we initialize a practical algorithm, called Greedy-conservative Q-ensemble learning (GCQL) that builds on top of State-of-the-Art offline RL and online RL method.
- We empirically verify the effectiveness of our method by comprehensive experiments on the popular continuous control tasks MuJoCo (Todorov et al., 2012) with offline dataset coming from D4RL (Fu et al., 2020).

## 2  RELATED WORK

**Online RL** In general, online RL algorithms can be divided into two categories, i.e., on-policy and off-policy algorithms. On-policy methods (Schulman et al., 2015; 2017) update the policy using data collected by its current behavior policy. As ignoring the logged data collected by its history behaviour policies, they usually have a lower sample efficiency than the off-policy RL. On the other hand, off-policy methods (Fujimoto et al., 2018; Chen et al., 2021) enable the policy to learn from experience collected by history behavior polices, however, they cannot learn well from history trajectories collected by other agents' behavior policies (Fujimoto et al., 2019; Kumar et al., 2020). Consequently, the need for huge online interaction makes online RL impractical for some real-world applications, such as dialog agents (Jaques et al., 2019) or education system (Mandel et al., 2014).

**Offline RL** Offline RL algorithms assume the online environment is unavailable and learn policies only from the pre-collected dataset. As the value estimation error cannot be corrected using online interactions here, these methods tend to utilize a pessimistic updating strategy to relieve the distribution mismatch problem (Fujimoto et al., 2019; Kumar et al., 2019). To implement such a strategy,

model-free offline RL methods generally employ value or policy penalties to constrain the updated policy close to the data collecting policy (Wu et al., 2019; Kumar et al., 2020; Fujimoto et al., 2019; He & Hou, 2020). And model-based methods use predictive models to estimate uncertainties of states and then update the policy in a pessimistic way based on them (Kidambi et al., 2020; Yu et al., 2020). Those offline RL methods cannot guarantee a good performance, especially when the data quality is poor (Kumar et al., 2020). Besides, policy evaluation when the online environment is unavailable is also challenging. Even though off-policy evaluation (OPE) methods (Dann et al., 2014) present alternative solutions, they are still far from perfect.

Above issues of online and offline RL motivate us to investigate the offline-online setting.

**Offline-online RL** Lee et al. (2021) and Nair et al. (2020) focus on the mixed setting where the agent is first learned from the offline dataset, and then trained online. Nair et al. (2020) propose an advantage-weighted actor-critic (AWAC) method that restricts the policy to select actions close to those in the offline data by an implicit constraint. When online interactions are available, such conservative constraint may have adverse effects on the performance. Lee et al. (2021) employ a balanced replay scheme to address the distribution shift issue. It uses the offline data by only selecting near-on-policy samples. Unlike these two works, *our method utilizes all online and offline data, and explicitly considers the difference between them by adaptively applying non-conservative or conservative updating schemes, respectively*. Matsushima et al. (2021) focuses on optimizing deployment efficiency, i.e., the number of distinct data-collection policies used during learning, by employing a behavior-regularized policy updating strategy. Although in terms of deployment efficiency, their work is between online and offline RL, it ignores existing offline dataset, and dose not focusing on improving sample efficiency, while both are addressed in our paper. Some works (Zhu et al., 2019; Vecerik et al., 2017; Rajeswaran et al., 2018; Kim et al., 2013) can also learn from online interactions and offline data. However, they need expert demonstrations instead of any dataset, and this limits their applicability.

## 3 PRELIMINARIES

In RL, the interaction between the agent and environment is usually modelled using Markov decision process (MDP) $(\mathcal{S}, \mathcal{A}, p_M, r, \gamma)$, with state space $\mathcal{S}$ (state $s \in \mathcal{S}$), action space $\mathcal{A}$ (action $a \in \mathcal{A}$). At each discrete time step, the agent takes an action $a$ based on the current state $s$, and the state changes into $s'$ according to the transition dynamics $p_M(s' \mid s, a)$, and the agent receives a reward $r(s, a, s') \in \mathbb{R}$. The agent's objective is to maximize the return, which is defined as $R_t = \sum_{i=t+1}^{\infty} \gamma^i r(s_i, a_i, s_{i+1})$, where $t$ is the time step, and $\gamma \in [0, 1)$ is the discounted factor. The mapping from $s$ to $a$ is denoted by the stochastic policy $\pi : a \sim \pi(\cdot|s)$. Policy can be stochastic or deterministic, and we use the stochastic from in this paper for generality. Each policy $\pi$ have a corresponding action-value function $Q^\pi(s, a) = \mathbb{E}_\pi[R_t \mid s, a]$, which is the expected return following the policy after taking action $a$ in state $s$. The policy $\pi$'s action-value function can be updated by the Bellman operator $\mathcal{T}^\pi$:

$$\mathcal{T}^\pi Q(s, a) = \mathbb{E}_{s'}[r + \gamma Q(s', \pi(s'))] \tag{1}$$

Q-learning (Sutton & Barto, 2011) directly learns the optimal action-value function $Q^*(s, a) = \max_\pi Q^\pi(s, a)$, and such Q-function can be modelled using neural networks (Mnih et al., 2015).

In principle, off-policy methods, such as Q-learning, can utilize experiences collected by any policies, and thus they usually maintain a replay buffer $\mathcal{B}$ to store and repeatedly learn from experiences collected by behavior policies (Agarwal et al., 2020). Such capability also enables off-policy methods to be used in the offline setting, by storing offline data into the buffer $\mathcal{B}$, and not updating the buffer during learning since no further interactions are available here (Levine et al., 2020). But this simple adjusting cannot guarantee the agent to have a reasonable performance, especially when the dataset is not diverse (Kumar et al., 2020; Agarwal et al., 2020; Fujimoto et al., 2019), and this is also the problem tackled in most offline RL works.

In this paper, *we focus on the offline-online setting, where the agent is first learned from the offline dataset, and then trained via online interactions*. And without additional remarks, online RL methods refer to off-policy algorithms in the rest of this paper. We only use off-policy methods because they can make more use of offline data than on-policy ones for gaining high sample efficiency, and on-policy methods are not compatible with our proposed framework introduced next.

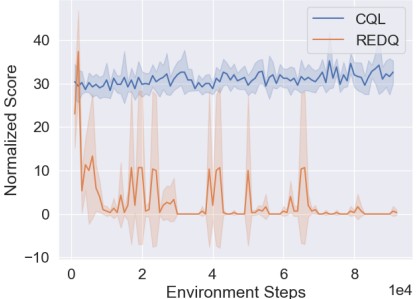

Figure 2: Learning curves on the D4RL (Fu et al., 2020) task Hopper-medium-replay-v0. The reported results are the averaged performance across five random seeds and the shaded areas represent the standard deviation across different seeds. The normalized score of 100 is the average returns of a domain-specific expert while normalized score of 0 corresponds to the average returns of an agent taking actions uniformly at random across the action space.

## 4 METHODOLOGY

In this section, we first give an illustrative example to explain our motivation. After that, we introduce the proposed general framework, namely Adaptive Q-learning, trying to couple online and offline RL in an organic way. To implement the proposed framework, then we present the Online-Offline Replay Buffer, targeting retaining and distinguishing online and offline data simultaneously. And finally we incorporate State-of-the-Art (SotA) online and offline RL algorithms into the framework, and introduce the proposed Greedy-Conservative Q-ensemble Learning algorithm in detail.

### 4.1 AN ILLUSTRATIVE EXAMPLE

We test the SotA offline RL method conservative Q-learning (CQL) (Kumar et al., 2020) and online RL method Randomized Ensembled Double Q-learning (REDQ) (Chen et al., 2021) under the offline-online setting. Here, we choose a widely used locomotion task Hopper (Todorov et al., 2012), and use the dataset hopper-medium-replay-v0 in the D4RL benchmark (Fu et al., 2020) as the offline dataset, which contains diverse experiences from different policies. Specifically, we first pre-train the agent with the offline dataset for $100K$ steps. Then, the agent is fine-tuned online by alternately conducting the interaction and updating process, where the agent interacts with the environment for $1K$ steps, and is updated for $10K$ steps. Such interleaving process ends until the total online interaction steps reach $90K$.

The offline-online setting in this task should be a favourable one for policy learning, because both diverse offline data and online interactions are available here. Therefore, we expect following results: the initial offline training using the dataset will provide a relatively good but not perfect starting point for the agent, and then the agent can be further improved by the online alternating process since online interaction data can be obtained, and finally we may acquire a well-performed policy, even with normalized score close to or better than 100 (i.e., the performance of an expert).

Nonetheless, experiment results are shown in Figure 2, and they do not totally meet our expectation. Specifically, starting points of two curves show that the offline algorithm CQL and online algorithm REDQ can both obtain normalized scores greater than 0 (i.e., the performance of a random policy) but not very high after the initial offline training process. This result is what we expected, because CQL is designed for the offline setting, and online off-policy methods, such as REDQ, can also learn from the diverse and large offline dataset even though it is fixed (Agarwal et al., 2020). However, although the starting scores (below 30) are far from the expert score and leave a large room for further promoting, both algorithms have troubles in the following online process. REDQ suffers from severe instability issue and its performance drops significantly during online learning, and in the end, the policy almost degenerates into a random one. One the other hand, even though the CQL agent can keep stable during learning, its improvement is very limited.

These results indicate that pure online RL algorithm may be problematic for effectively handling offline data and online interaction data in a single training process, and pure offline RL algorithm cannot make good use of valuable online interaction data due to its conservative updating strategy. Such observation motivates us to couple them in an organic way.

## 4.2 ADAPTIVE Q-LEARNING FRAMEWORK

In this subsection, we introduce our proposed framework. The underlying idea is simple and can be described as follows. When the agent learns from online and near-on-policy interaction data, we choose a more greedy or no-pessimistic updating strategy, since these data reflect the truth situation of the current policy. By contrast, when data are sampled from the offline dataset, we tend to use a more pessimistic updating strategy. Through such an adaptive way, we can make full use of both online and offline data, and explicitly consider their differences by separately applying suitable updating schemes.

Then, we give a formalization for above intuition based on Q-learning, and call this framework *Adaptive Q-learning*. The updating function of this framework can be defined by the following equation:

$$Q^{k+1} \leftarrow \arg\min_{Q^k} \left[ \mathbb{A}(Q^k) + \mathcal{W}(s,a)\mathbb{B}(Q^k) \right]. \tag{2}$$

This function consists of two terms. The first term $\mathbb{A}(Q)$ stands for the greedy updating strategy, which is a regular updating function for Q-value in online RL algorithms, e.g., the bellman error. The second term $\mathbb{B}(Q)$ stands for the pessimistic updating strategy, which is the value penalty, e.g., the Q-value regularizer (Wu et al., 2019). Besides, a weight function $\mathcal{W}(s,a)$ is applied to the penalty term $\mathbb{B}(Q)$. This weight function is based on the sampled data type. Specifically, when we use online and near-on-policy interaction data, $\mathcal{W}(s,a)$ will be a smaller value, and the updating relies more on the objective of online RL, leading to a relatively greedy strategy. On the contrary, when we use offline data, $\mathcal{W}(s,a)$ will be a bigger value, and the updating strategy is relatively pessimistic.

**Remark:** Our behind intuition can also be easily formalized via the policy learning objective. Such variant of our framework can be denoted by $\pi^{k+1} \leftarrow \arg\max_{\pi^k} \left[ \mathbb{A}\left(\pi^k\right) + \mathcal{W}(s,a)\mathbb{B}\left(\pi^k\right) \right]$, where $\mathbb{A}(\pi^k)$ is a objective for the policy in online RL and $\mathbb{B}(\pi^k)$ is a policy penalty term. We also provide a implementation for this variant based on TD3+BC algorithm (Fujimoto & Gu, 2021), and our framework can largely boost its performance within limited environment steps in most tasks (see Appendix D for details).

## 4.3 OORB: ONLINE-OFFLINE REPLAY BUFFER

Next, we introduce a simple but effective online-offline replay buffer (OORB) to distinguish between near-on-policy online interaction data, and the offline data. OORB consists of two replay buffers. One is the online buffer that collects the online interaction data. Besides, to ensure the data in the online buffer is near-on-policy, we set it to be very small, and fresh online interaction data are stored into it by following a first-in-first-out rule. The other is the offline buffer consisting of the newly generated online interaction data and the offline dataset which may come from any policies.

Data are sampled from OORB following a Bernoulli distribution, which means that with a probability $p$, they are sampled from the online buffer, and with probability $1 - p$, they are sampled from the offline buffer. To benefit from both online and offline data in a balanced way, we empirically set $p$ to 0.5, and its effect on the final performance is further tested via ablation studies in Section 5.4. Results show that $p = 0.5$ works best overall, which confirms our claim that offline data and online interaction data are both crucial for policy learning.

## 4.4 GCQL: GREEDY-CONSERVATIVE Q-ENSEMBLE LEARNING

We then present a detailed implementation of our proposed framework, by incorporating SotA offline RL algorithm CQL and online RL algorithm REDQ, and we call our implemented algorithm *Greedy Conservative Q-ensemble Learning* (GCQL). Specifically, we use the updating function in REDQ as the first term $\mathbb{A}(Q)$ in Equation 2, and use the conservative regularizer in CQL as the

---

**Algorithm 1:** Greedy-conservative Q-ensemble learning

---
1  **Initialization**:
2  Initialize policy $\pi_\phi$, ensemble Q functions $Q_{\theta_i}, i \in N$, offline training steps t
3  Online exploration steps $T_{on}$, offline update steps $T_{off}$
4  Initialize $T_{initial}$, sample possibility $p$, start sampling steps: $T_s$
5  Initialize online buffer $B_{\text{on}}$ to empty, offline buffer $B_{\text{off}} \leftarrow$ offline dataset
6  Initialize online buffer size $S_{\text{on}} \leftarrow 0$
7  **Initial offline learning**:
8  Train the agent for $T_{initial}$ steps using the logged dataset
9  **while** *True* **do**
10    $t \leftarrow 0$
11    Explore $T_{on}$ steps online
12    Store the $T_{on}$ steps experiences to both online buffer $B_{\text{on}}$ and offline buffer $B_{\text{off}}$
13    $S_{on} \leftarrow S_{on} + T_{on}$
14    **for** $t < T_{off}$ **do**
15       Sample a random value $p_s \sim \mathbb{U}(0,1)$
16       **if** $p_s < p$ *and* $S_{on} > T_s$ **then**
17          Sample a batch $(s, a)$ from online buffer $B_{\text{on}}$
18          Set the $\mathcal{W}(s,a)$ to 0
19       **end**
20       **else**
21          Sample a batch $(s, a)$ from offline buffer $B_{\text{off}}$
22          Set the $\mathcal{W}(s,a)$ to 1
23       **end**
24       Update the Q functions by Equation 3
25       Update the policy by Equation 5
26       t += 1
27    **end**
28 **end**

---

second term $\mathbb{B}(Q)$, and the updating function can be presented by the following equation:

$$
\begin{aligned}
Q_i^{k+1} \leftarrow \arg\min_{Q_i^k} \Bigg\{ &\frac{1}{2}\mathbb{E}_{\mathbf{s},\mathbf{a},\mathbf{s}' \sim \mathcal{D}_{\text{OORB}},\mathbf{a}' \sim \pi_\phi(\cdot|s')} \left[ \left( Q_i^k(\mathbf{s},\mathbf{a}) - \hat{\mathcal{B}}^\pi \hat{Q}^k(\mathbf{s}',\mathbf{a}') \right)^2 \right] \\
&+ \mathcal{W}(\mathbf{s},\mathbf{a})\alpha \mathbb{E}_{\mathbf{s} \sim \mathcal{D}_{\text{OORB}}} \left[ \log \sum_{\dot{\mathbf{a}}} \exp(Q(\mathbf{s},\dot{\mathbf{a}})) - \mathbb{E}_{\mathbf{a} \sim \mathcal{D}_{\text{OORB}}}[Q(\mathbf{s},\mathbf{a})] \right] \Bigg\}
\end{aligned}
\tag{3}
$$

where the action $\dot{\mathbf{a}}$ is sampled from current policy, i.e., $\dot{\mathbf{a}} \sim \pi_\phi(\cdot|s), \mathbf{s} \sim \mathcal{D}_{\text{OORB}}$ and $\mathcal{D}_{\text{OORB}}$ is the OORB replay buffer. $\hat{\mathcal{B}}^\pi \hat{Q}^k(\mathbf{s},\mathbf{a})$ is defined by

$$
r + \gamma \min_{i \in \mathcal{M}} \hat{Q}_i^k(\mathbf{s}',\mathbf{a}'), \quad \mathbf{a}' \sim \pi_\phi(\cdot \mid \mathbf{s}').
\tag{4}
$$

We randomly select two Q functions from the ensemble Q functions and the $\mathcal{M}$ represents the selected Q functions' index just following the REDQ's setting. The $\hat{Q}$ stands for a target $Q$ function for stabilizing the learning process (Mnih et al., 2015). The update function of policy is defined by:

$$
\pi_{\phi_{k+1}} \leftarrow \arg\max_{\pi_{\phi_k}} \mathbb{E}\left[ \mathbb{E}_{i \in N}\left[Q_i(\mathbf{s},\mathbf{a})\right] - \alpha \log \pi_{\phi_k}(\mathbf{a} \mid \mathbf{s}) \right], \quad \mathbf{a} \sim \pi_{\phi_k}(\cdot \mid \mathbf{s})
\tag{5}
$$

When sampled data is from the online replay buffer, we set $\mathcal{W}(s,a)$ to 0, otherwise 1, and this is why we use the term "greedy-conservative" to describe our algorithm. In another word, we greedy exploit the near-on-policy online data by the regular online RL scheme without any conservative regularizer. On the contrary, we conservatively exploit the offline data by employing the offline RL

| Environment | GCQL (Ours) | CQL | REDQ | AWAC |
|---|---|---|---|---|
| walker2d-random | **53±27** | 16±9 | 19±3 | 12 |
| hopper-random | **84±40** | 11±1 | 12±17 | 63 |
| halfcheetah-random | **100±2** | 46±4 | 34±1 | 53 |
| walker2d-medium | **94±6** | 83±1 | 5±3 | 80 |
| hopper-medium | **105±1** | 100±1 | 3±1 | 91 |
| halfcheetah-medium | **66±3** | 42±0 | 46±1 | 41 |
| walker2d-expert | **117±2** | 113±1 | 6±1 | 103 |
| hopper-expert | **114±1** | 113±1 | 12±6 | 112 |
| halfcheetah-expert | **110±0** | 108±1 | 1±0 | 106 |
| walker2d-medium-expert | **117±4** | 113±1 | 12±3 | 78 |
| hopper-medium-expert | **115±1** | 114±0 | 40±15 | 112 |
| halfcheetah-medium-expert | **107±1** | 95±2 | 9±3 | 41 |
| walker2d-medium-replay | **114±6** | 64±5 | 53±11 | - |
| hopper-medium-replay | **96±9** | 39±4 | 38±9 | - |
| halfcheetah-medium-replay | **59±2** | 50±0 | 50±1 | - |

Table 1: Performance of policies trained using 90K online interaction steps for GCQL, CQL and REDQ, and 500K online interaction for AWAC whose results are taken from Nair et al. (2020).± captures the standard deviation over seeds. The reported results are the average test performance across five random seeds in our experiments. The learning curves are showed in Appendix A.

regularizer. Formally, this strategy can be explained by Definition 6:

$$\mathcal{W}(s,a) \leftarrow \begin{cases} 0 & \text{if } (s,a) \text{ is sampled from the online replay buffer} \\ 1 & \text{otherwise} \end{cases} \qquad (6)$$

Algorithm 1 summarizes our proposed method. And we also explain the main steps in the Algorithm as follows. Firstly, We first learn from the existing offline data for $T_{initial}$ steps to leverage them. To make good use of the offline data, we usually set a big value for $T_{initial}$, e.g., $100K$. Secondly, We begin the following interleaving learning steps. We conduct the online exploration process for $T_{on}$ steps and store the new experiences to OORB. We set the online exploration steps $T_{on}$ to a small value, e.g., $1K$. In contrast, the offline update step $T_{off}$ is set to be larger than $T_{on}$, e.g., $10K$. We sample a batch from our OORB and update the policy and Q-functions. If the sampled batch comes from the online buffer, then we set the weight value $\mathcal{W}(s,a)$ to 0, otherwise 1. The above interleaving learning process is repeated till the end.

## 5 EXPERIMENTS

In this section, we design experiments to verify the effectiveness of our method from three perspectives: (1) the performance superiority compared with other baselines; (2) ablation studies to test the effect of each component used in our method. (3) the influence of different hyper-parameters.

### 5.1 SETTINGS

All experiments were done on the continuous control benchmark MuJoCo (Todorov et al., 2012), and the offline dataset comes from the popular offline RL benchmark D4RL (Fu et al., 2020). To make interaction limited, we set the number of online interaction steps for each iteration to a small value, i.e., $1K$. To better exploit the offline dataset, we set the number of offline training steps to a large value, i.e., $100K$ for $T_{intial}$, and $10K$ for $T_{off}$. The training process ends until the number of all online interaction steps reach $90K$, and the number of all offline updating steps reach $1M$. For OORB, we set $p = 0.5$ as described in Section 4.3. The size of the online buffer is set to $20K$, and the size of the offline buffer is set to $3M$. For other hyper-parameters, we follow the default setting in baselines, except for the number of the ensemble Q, which is 5 in our experiments. The above configurations keep same across all tasks. As our main purpose is to present a new framework instead of gaining SotA performance, we do not fine-tune these hyper-parameters for each task.

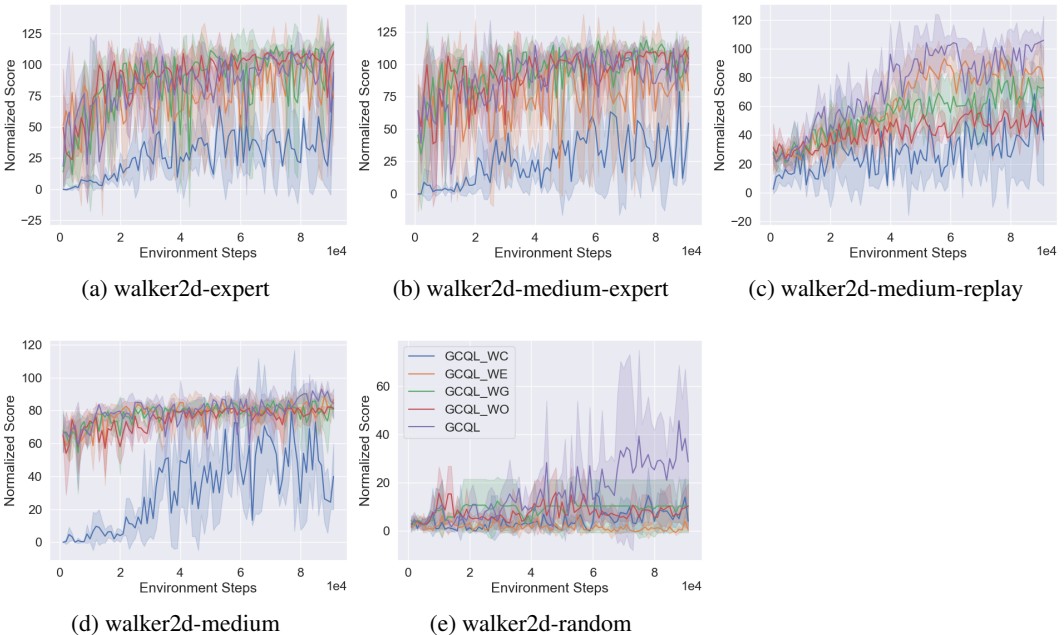

Figure 3: Ablation study on Walker2d task.

## 5.2 OVERALL PERFORMANCE

4 methods, i.e., GCQL (Ours), CQL, REDQ and AWAC, are tested on 3 tasks, i.e., Walker, Hopper and HalfCheetah, and each task has 5 different kinds of offline dataset, which are random-v0, medium-v0, expert-v0, medium-expert-v0, and medium-replay-v0. As online interactions are available, we take the maximization over testing scores during the whole training process, and report the average of these max scores across five different seeds in Table 1. Please note that the results of AWAC are directly taken from their paper (Nair et al., 2020). As shown in Table 1, our method gains a better performance than the baselines in all tasks. Particularly, for the medium-replay and random dataset, our GCQL outperforms the baselines by a large margin. By contrast, CQL also achieves an expert performance on the high-quality dataset, e.g., expert and medium-expert tasks. For these high-quality datasets, our method achieves a comparable or slightly better performance over CQL. In terms of the online RL method REDQ, it fails on almost all tasks. Even though we employ an online limited exploration process, conventional online RL cannot benefit from such limited experiences. On the contrary, such limited online experiences could bring catastrophic consequences, e.g., on the halfcheetah-medium-replay task, REDQ suffers a significant instability issue. On the other hand, when dataset quality is high, e.g., expert or medium-expert dataset, our method can leverage the benefit of the offline RL that gains an expert performance with very limited interaction steps, such as around $20K$ steps. By contrast, when dataset quality is poor, e.g., the random or medium-replay dataset, only our method can learn effectively with less than $100K$ online steps, which demonstrate that our method can take the advantage of the limited online experiences as much as possible. *In sum, our method can benefit the most from the limited online experience while still maintain the offline learning's ability.* Besides, We also include the corresponding learning curves in Appendix A for a comprehensive understanding. These learning curves demonstrate that our method also achieves a better final average score at the last iteration in most cases.

## 5.3 ABLATION STUDIES

To investigate each component's effect in our method, we conduct the following ablation studies. To do this, we design four variants of our method. **GCQL_WE**: GCQL without the ensemble where we only use two Q functions same as the setting of CQL. **GCQL_WO**: GCQL without the online replay buffer where we only sample from the offline buffer. **GCQL_WG**: GCQL without the greedy

strategy where we fix the weighed $\mathcal{W}$ to 1. **GCQL_WC**: GCQL without the conservative term where we fix the $\mathcal{W}$ to 0.

We test all 5 kinds of offline dataset for the task Walker2d, and results averaged over three different random seeds are shown in Figure 3. First, from Figure 3, it is easy to deduce that when the quality of data is high, i.e., including the expert dataset, almost all variants perform well except the GCQL_WC. The underlying reason is obvious as we firstly pre-train the policy by offline data. If no such conservative restriction, then the pre-trained policy would suffer a serious distribution mismatch issue (Kumar et al. (2020)) and gain a poor pre-trained policy at last. By contrast, with the conservative scheme, a relatively good pre-trained policy can be obtained. That is why the starting point of GCQL_WC is much lower than other variants.

Secondly, when the dataset is diverse, which means that the dataset is collected by different behavior policies and includes data from different distributions, such as the medium-replay dataset, the greedy scheme (including the offline-online replay buffer and the greedy update strategy) plays an important role while the ensemble feature seems have limited influence. For instance, the performance declined significantly for GCQL_WO and GCQL_WG on walker2d-medium-replay.

Thirdly, when the dataset is not diverse, such as the medium dataset which is collected by one medium policy, almost all curves grow slowly except GCQL_WC. This may be caused by the characteristic of the dataset, because dataset diversity plays an important role for policy learning (Agarwal et al., 2020). However, according to the learning curves, our method still has a clear performance improvement at the latter learning stage. For the random dataset, all variants fail due to the poor quality of the dataset. Instead, our method achieves a clear performance improvement under it, which indicates that every component is important in this case.

## 5.4 ANALYSIS ON HYPER-PARAMETERS

As we do not fine-tune the hyper-parameters in our experiments, one may wonder how the hyper-parameters affect the performance. To this end, we conduct experiments to investigate their influence. The detailed learning curves are shown in Appendix B. As one main characteristic of our method is limited interaction, we fix the online exploration step as $1K$ for each iteration in the MuJoCo benchmark. We then try settings with different initial update steps $T_{initial}$, offline update steps $T_{off}$, and sample possibility from online buffer $p$. Specifically, $T_{initial}$ is tested with $2e5$ and $5e4$; $T_{off}$ is tested with $2e4$ and $5e3$; $p$ is tested with $0.3, 0.4, 0.5, 0.6$ and $0.7$.

According to the evaluation results, we may conclude that the performance of our method is insensitive to the $T_{initial}$ and $T_{off}$, especially for datasets with not poor quality, e.g., datasets except for the random one. By contrast, $p$ has a bigger impact on the performance. Particularly, methods with higher $p$ performs better on the medium and medium-replay tasks, while the ones with lower $p$ perform better on the other tasks. That indicates that when dataset quality is very good or very bad, methods with more offline updating perform better. On the other hand, only the variant of $p = 0.5$ can achieve a clear performance improvement in both the medium and random datasets. Overall, the default setting $p = 0.5$ is the most suitable setting, which indicates that taking the online and offline data equally important may be the best option in most cases.

## 6 CONCLUSIONS

This paper first discusses the disadvantages of online and offline RL, and the shortcomings of current offline-online RL methods. Then, considering that offline and near-on-policy online datasets are both crucial for policy learning, we propose a unified framework that can adaptively and effectively take advantage of both offline and online data. Furthermore, a practical algorithm based on the framework that greedily exploits the online experiences and conservatively exploits the offline experiences is presented. We conduct comprehensive experiments to verify the effectiveness of our method. In terms of the shortcomings, although our framework can take advantage of the offline and limited online dataset, the quality of the offline dataset still has a big impact on the performance. For instance, our method is relatively slower when learning from the random dataset. At last, we hope this work could contribute to bridging the gap between the practice and DRL research.

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

## A  LEARNING CURVES FOR ALL TASKS

Figure 4 indicate the whole learning curves for GCQL, REDQ and CQL.

## B  EXTRA EXPERIMENTS ON HYPER-PARAMETERS

Figure 5 and 6 shows the different hyper-parameters' setting on the performance.

## C  EXTRA ABLATION STUDY ON HALFCHEETAH

Figure 7 present extra ablation study on halfcheetah. GCQL_WG is the GCQL without the greedy update scheme but still with the online-offline two-level buffer, while GCQL_WGO is the GCQL without the greedy scheme and online-offline replay buffer. From Figure 7 c,d,e, it is clear that the greedy scheme plays an important role in boosting the performance when the dataset is not optimal or near-optimal. On the other hand, from Figure 7 d, we can see the online-offline replay buffer is crucial for stabilizing the learning process where the GCQL_WGO and CQL suffer serious stability issues which may be caused by extrapolation error (Fujimoto et al., 2019).

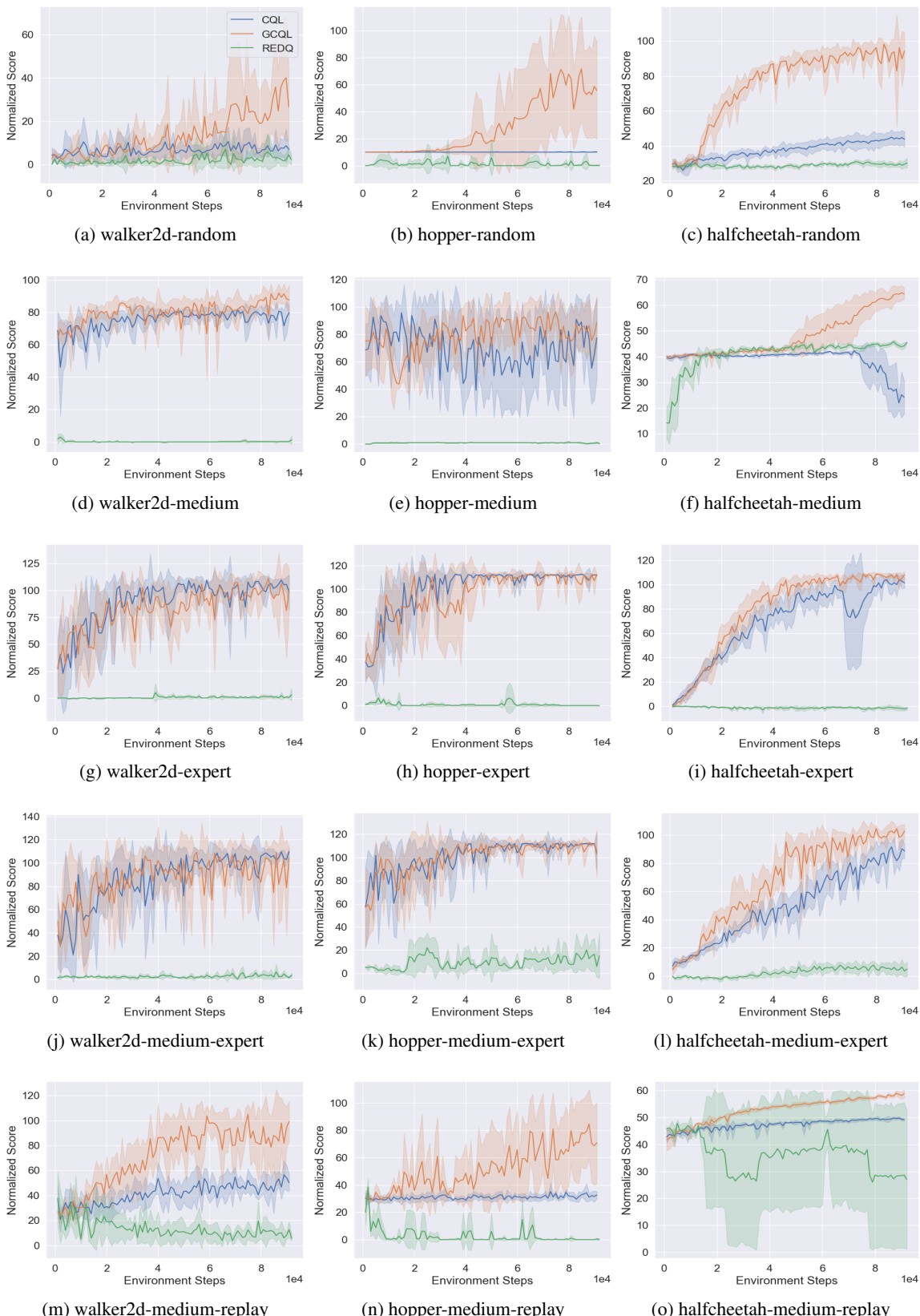

Figure 4: Training curves on D4RL continuous control benchmark across five random seeds. The shaded areas represent the standard deviation across different seeds.

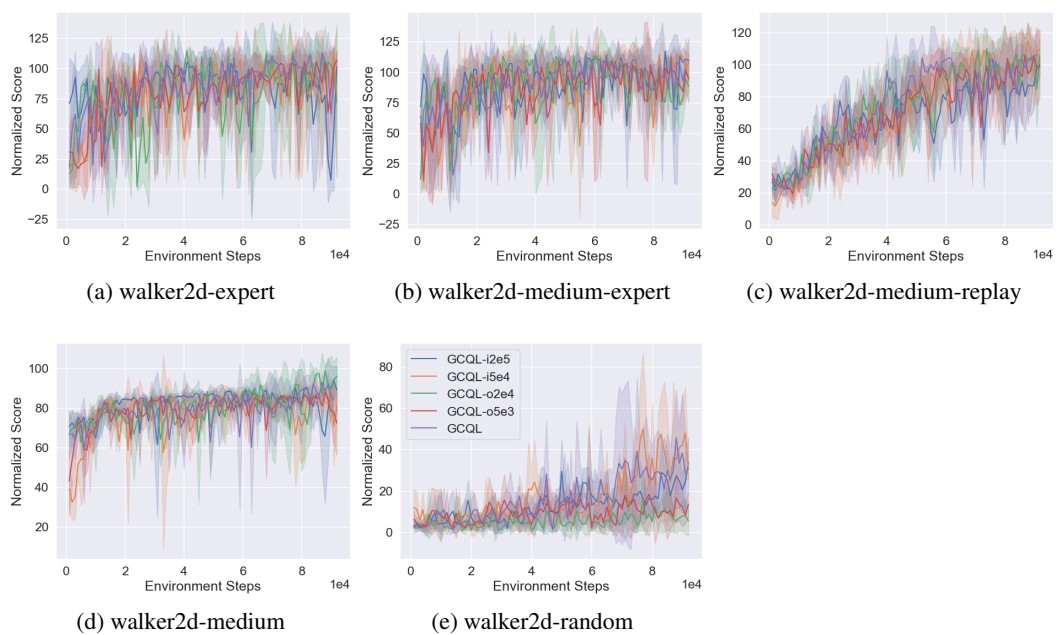

(a) walker2d-expert       (b) walker2d-medium-expert       (c) walker2d-medium-replay

(d) walker2d-medium       (e) walker2d-random

Figure 5: Different offline update steps on walker2d task across three random seeds.GCQL-i2e5:initial steps is 2e5, GCQL-i5e4: initial stesp is 5e4, GCQL-o2e4: offline update steps is 2e4, GCQL-o5e3: offline update steps is 5e3.

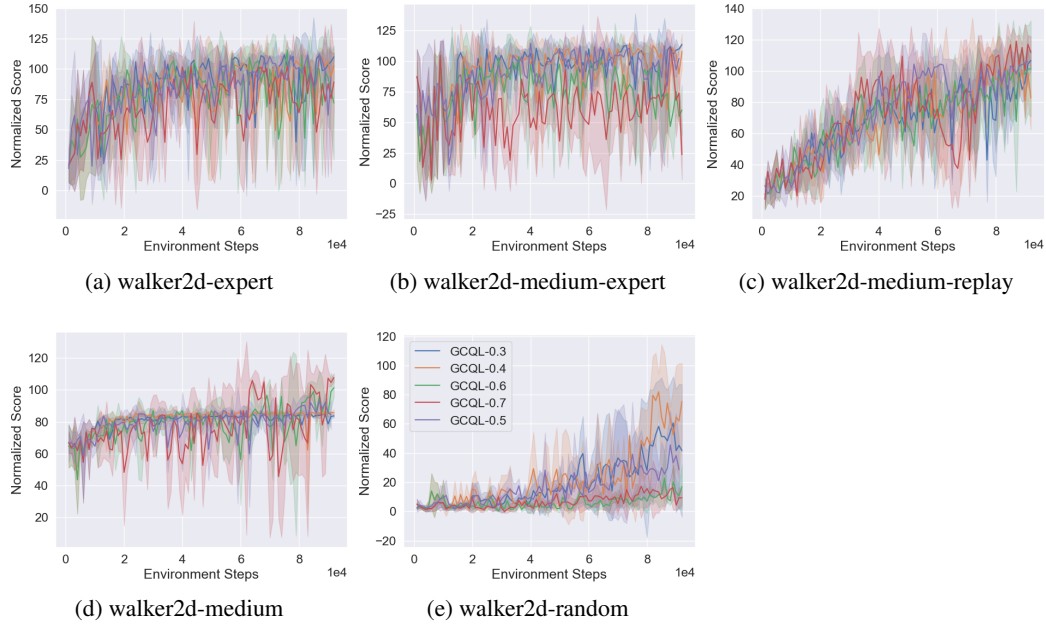

(a) walker2d-expert       (b) walker2d-medium-expert       (c) walker2d-medium-replay

(d) walker2d-medium       (e) walker2d-random

Figure 6: Different possibility setting on walker2d task across three random seeds.GCQL-X means the sample possibility from online buffer is X.

# D  EXTRA EXPERIMENTS ON TD3+BC

Extra experiments on TD3+BC (Fujimoto & Gu, 2021) (without states normalization) for the sub-optimal dataset, i.e., random, medium, and medium-replay, as the offline RL method can per-

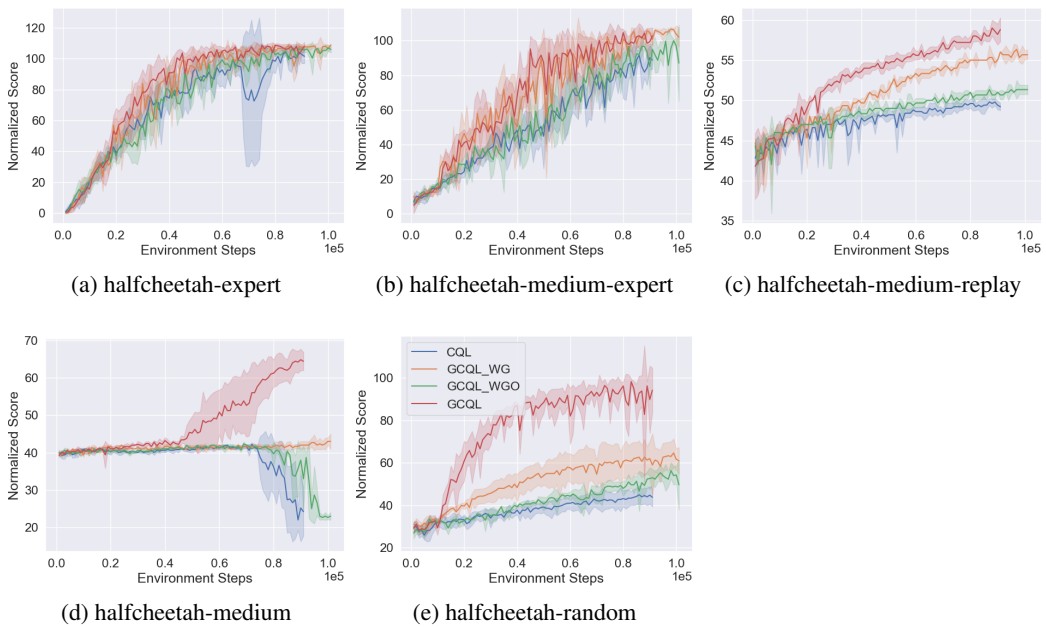

Figure 7: Extra ablation study on Halfcheetah.

form very well on expert or near-on-expert datasets. We simply apply our greedy-conservative learning framework and online-offline buffer to TD3+B without any fine-tuning and call it: TD3BCGC. The learning curves indicate that our greedy-conservative framework can still improve policy-penalty-based methods by a large margin in most tasks. Moreover, it is very simple to apply our framework to TD3+BC where less than 20 lines codes are needed. We modify the policy objective from $\pi = \text{argmax}_\pi \mathbb{E}_{(s,a)\sim\mathcal{D}} \left[ \lambda Q(s, \pi(s)) - (\pi(s) - a)^2 \right]$ to $\pi = \text{argmax}_\pi \mathbb{E}_{(s,a)\sim\mathcal{D}} \left[ \lambda Q(s, \pi(s)) - \mathcal{W}(s, a)(\pi(s) - a)^2 \right]$, where $\mathcal{W}(s, a)$ follows our setting introduced in section 4.4.

# E    EXTRA EXPERIMENTS ON OFF2ON

Here, we compared our method with OFF2ON (Lee et al., 2021) which also employs an online update scheme to fine-tune a pre-trained agent. As the author did not provide the pre-training code for CQL, we use the pre-trained CQL agents provided by OFF2ON's author for online fine-tuning. All the hyper-parameters follow the default setting except the online interaction steps. Figure 9 indicated the whole learning curves. From this figure, we can see GCQL has a comparable or better performance than OFF2ON. It is worth noting that OFF2ON fine-tuning their method, for instance, the critic's neural network architecture is different from the original CQL paper. In contrast, we did not fine-tune these parameters, and all settings are the same for all tasks.

# F    EXTRA EXPERIMENTS ON OFFICIAL ONLINE REDQ

In this section, we conduct extra experiments on the online REDQ (Chen et al., 2021). To guarantee the reproduction performance, we use the official code from the author, and all hyper-parameters are following its default setting, e.g., the number of Q is 10, and the utd-ratio is 20. In contrast, the number of Q in GCQL is 5, and utd-ratio is 10. REDQ-ONLINE updates its policy following a conventional online RL setting, i.e., learning from scratch without the offline pre-training. From Figure 10, it is clear that GCQL achieves a better or comparable performance than REDQ-ONLINE except for the walker2d-random dataset. Specifically, when the dataset is good, e.g., including the expert dataset, GCQL can outperform the REDQ-ONLINE by a large margin. On the other hand,

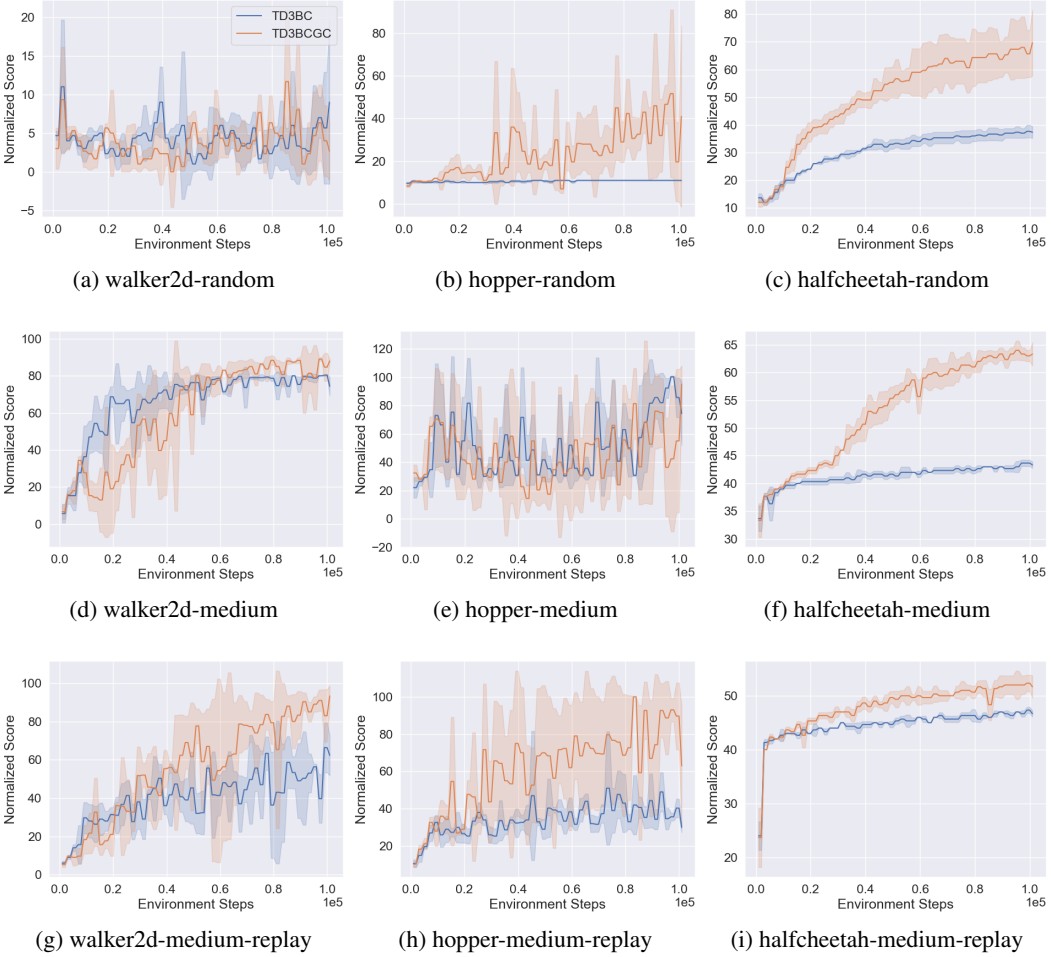

Figure 8: Training curves for TD3+BC on D4RL continuous control benchmark across three random seeds on tasks: random-v0, medium-v0 and medium-replay-v0. TD3BC means the algorithm introduced by (Fujimoto & Gu, 2021) while TD3BCGC is the variant with our greedy-conservative framework and online-buffer replay buffer.

when the offline dataset is of poor quality, i.e., random-dataset, GCQL can still learn a comparable or better policy than REDQ-ONLINE in two of three tasks, i.e., the hopper-random and halfcheetah-random.

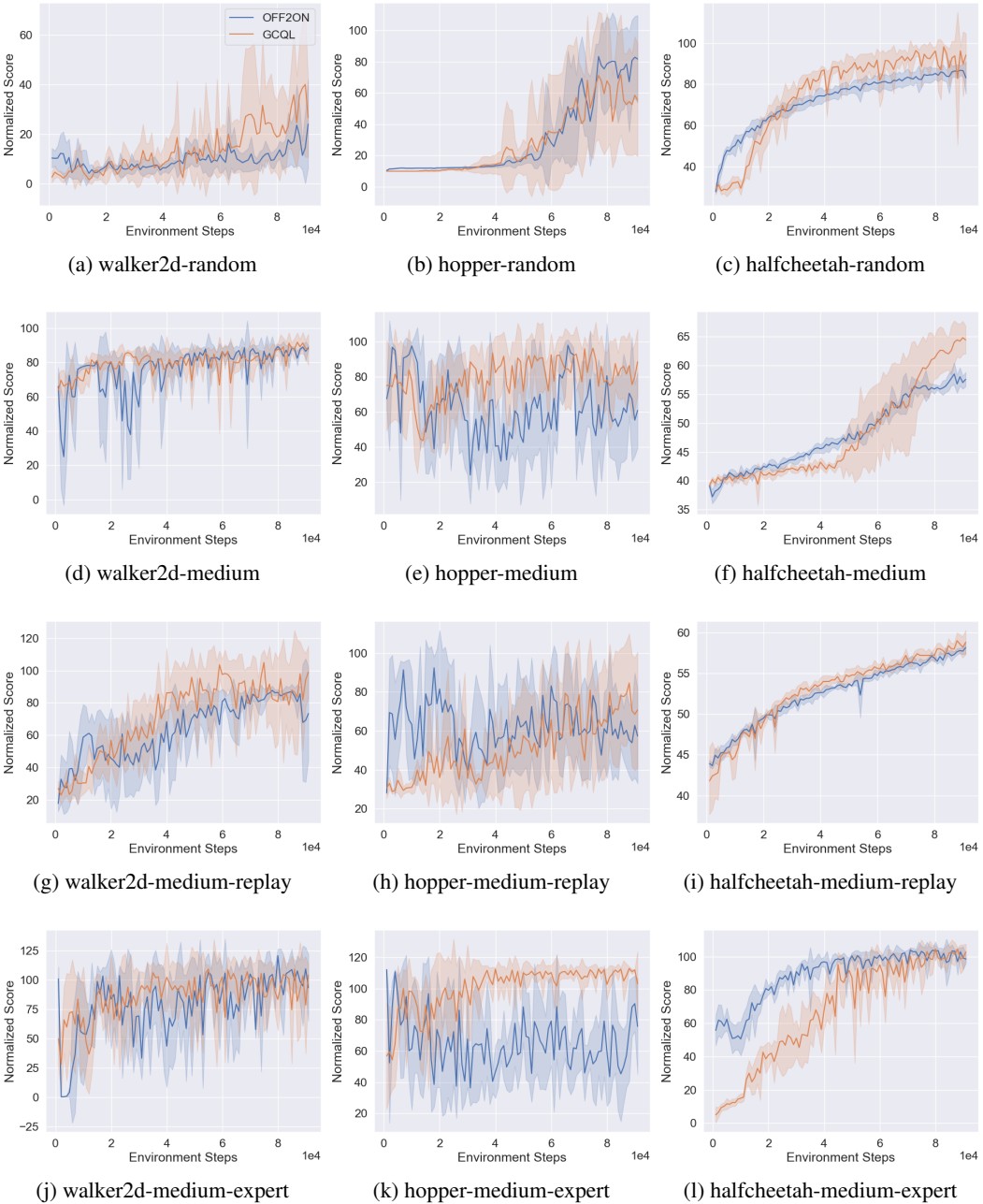

Figure 9: Training curves for OFF2ON (Lee et al., 2021) on D4RL continuous control benchmark across four random seeds on tasks: random-v0, medium-v0, medium-expert-v0 and medium-replay-v0.

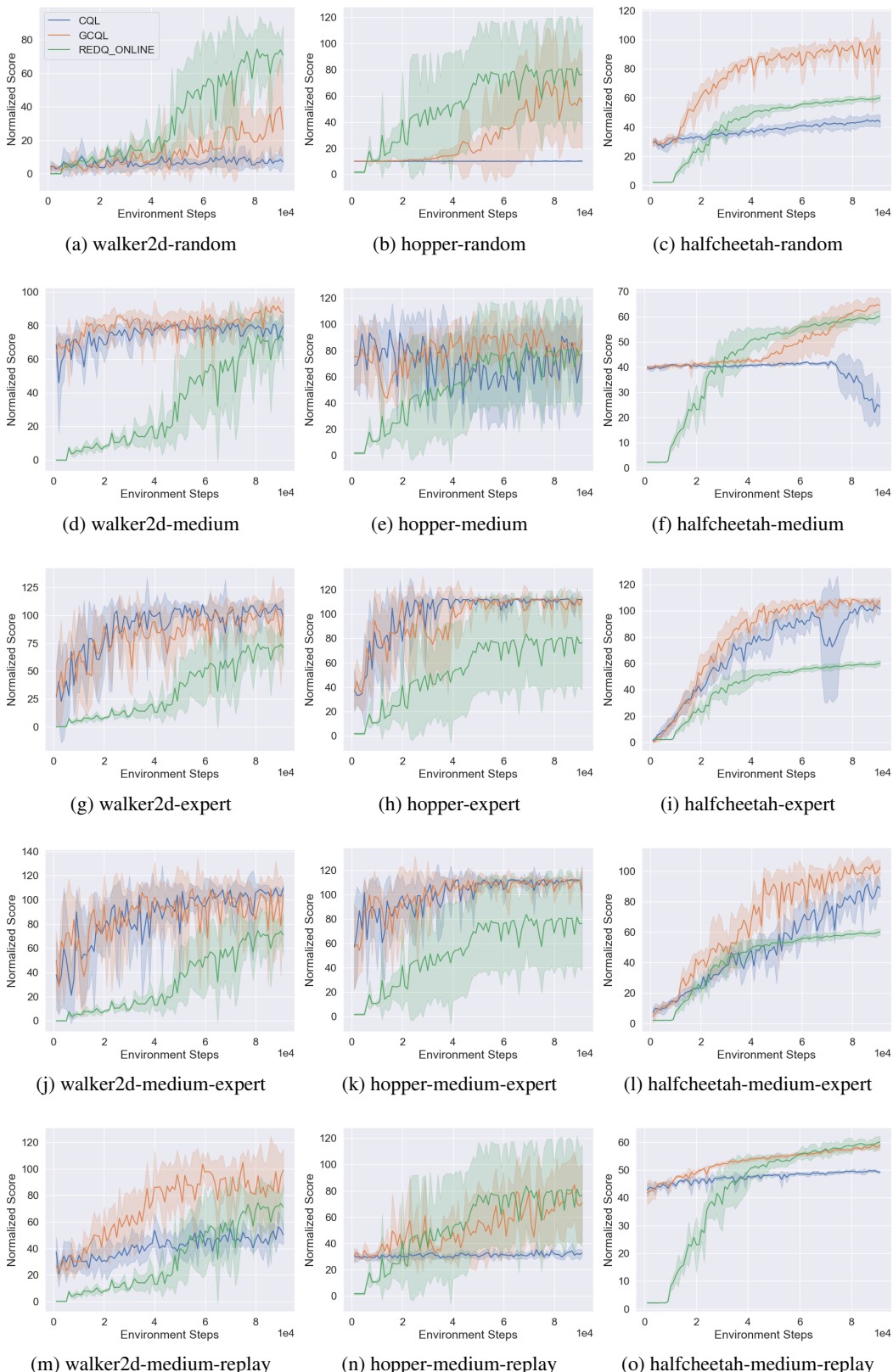

Figure 10: REDQ-ONLINE learn from scratch without the offline pre-training while GCQL and CQL learn from both offline dataset and online interaction.

