# OpenReview forum: "Adaptive Q-learning for Interaction-Limited Reinforcement Learning"
_ICLR.cc/2022/Conference — ICLR 2022 Submitted_

### Official Review · Reviewer_VmYw · 2021-11-03

**Correctness:** 3
**Technical Novelty And Significance:** 3
**Empirical Novelty And Significance:** 2
**Recommendation:** 6
**Confidence:** 4

**Main Review:**


Main points:
- The topic is interesting and I find that combining both offline and online training is a fruitful direction of research, which is potentially relevant in many practical situations.
- The main intuition for adaptive Q-learning make sense to me---treating the offline and online data differently, with modified updates for each.
- In terms of clarity, some parts need to be explained further, such as how the algorithm uses the replay buffers.  See the section below for more detailed questions.
- The experiments seemed to be done well generally and the performance of the proposed algorithm is substantially better than the baselines in certain settings e.g. medium-replay. It's also interesting to have experiments where the online interactions with the environment are limited. Other ablation experiments and sensitivity tests to hyperparameters are included, which are welcome additions. The ablation studies seem to indicate that the proposed algorithm could be simplified though and I've included a few questions below.

Questions and suggestions:
- The description of the online-offline replay buffers and section 4.3 are unclear. What exactly does each replay buffer contain? The online replay buffer says it contains "online exploration experience" but then the same thins is said about the offline buffer, "offline replay buffer consisting of the newly generated online exploration data". I'm confused about this distinction. Does the offline replay buffer contain more than just the initial dataset?

- Why is it that REDQ fails completely on these tasks?
Other variants of Q-learning seem to work in batch RL such as the distributional RL algorithms or Random Ensemble Mixture (REM) from "An Optimistic Perspective on Offline Reinforcement Learning" (Agarwal et al.). REDQ seems to be fairly similar to REM so one could expect it to work decently well in the offline setting too.
Also, looking at the learning curves in the appendix, there doesn't seem to be any improvement even in the online part for REDQ.
Is the offline training a hindrance to the algorithm? Would it perform better if it was simply initialized from scratch without the offline training? I think this would be a useful baseline to include too.

- CGQL-wo "without the online replay buffer where we only sample from the offline buffer". Does this mean that all the experience gets pooled into a single replay buffer? Could clarify the mechanism?
Also, does that also include removing the conservative term? The full GCQL would set W=0 when sampling from the offline buffer.

- Related to the previous point, how important is it that the online buffer is much smaller than the offline buffer?

- To clarify, is the difference between REDQ and CGQL-wc (without the conservative term) only that the replay buffers are separated into two? Are all the other hyperparameters the same? Would this explain why REDQ fails but CGQL-wc can achieve decent performance?

- Do you have any idea what happened to CQL in halfcheetah-medium? In this environment, it seems like training more online actually leads to decreased performance over time, strangely enough.

- Looking at the ablations, it seems like the setting CGQL-wg does quite well overall. This would suggest that being more conservative the entire time is not a bad strategy. I'm a bit curious how these results compare to CQL.

- In the ablations, CGQL-we does about as well as the full CGQL in every setting except walker2d-random. In that case, even full GCQL doesn't improve that much over training, only reaching an average performance of around 30 by the end. This would suggest that the algorithm could be simplified while retaining almost identical performance. I'm not very convinced that adding ensembles
is crucial here.

- Section 5.4. The results should be included in the main text, even if it is summarized. Currently, there are no results to look at. Also, to present the different hyperparameter settings, I would just write that T_off was tested with 2e4 and 5e3 instead of describing the legend for the learning curves. Same for T_initial and p. In the current draft, the legend is out-of-place since the plot isn't in the main text. Also, I would add the label descriptions with the learning curves in the appendix for easier reading.

Minor points:
- "Moreover, how to evaluate the policy under the offline setting is also a challenging problem." While this is mentioned in the abstract, the paper does not tackle this problem so I think it would be best to remove it.

- I'm not sure if "framework" is the proper word to use to describe adaptive Q-learning. I usually expect "frameworks" to be more general and adaptive Q-learning seems like a fairly specific template for an algorithm. Personally, I found this a bit confusing and had different expectations when reading the abstract.

Various typos and awkward sentences are scattered throughout and should be proofread. I'm assuming these can be easily fixed so I haven't counted them as a negative point in my review. Some examples:
- p.1 "or by combination with the imitation learning" -> "or in combination with imitation learning"
- p.1 "pre-existing offline data can prevent agents
from converging prematurely due to the potential diverse, ...". Missing words?
- p.2 "a well-performed policy " -> "high-performing"/"strong"/"good"
- p.2 "In general, Online RL include the on-policy and off-policy RL" -> In general, online RL includes on-policy and off-policy RL"
- p.5 "CQL (Fujimoto et al 2019)". Wrong citation.
- p.5 "As explained in section 1, We take the online-offline buffer equally important". Awkward phrasing.


**Summary Of The Paper:**

This paper tackles a variation to the offline RL setting, where the agent is allowed some limited number of online interaction steps after learning offline. An algorithm, CGQL, is proposed for this setting and uses the idea that online and offline data should be used in different updates. Experiments on a common benchmark show that this approach can be more effective than standard batch RL methods and ablation studies are also included.


**Summary Of The Review:**

The problem the paper tackles is interesting, offline RL with limited online interactions. I think this is a nice direction of research that merits further investigation. The proposed approach makes intuitive sense to me and the algorithm, CGQL, has good performance on the benchmark environments. I do have a number of clarification questions and the paper would require some editing to make these clear. Overall, I am currently leaning towards acceptance.

---

> ### Author Response · Authors · 2021-11-18
> **Responses to Reviewer VmYw**
>
> Thank you for your valuable comments. Below is our response to your concerns.
> >"The description of the online-offline replay buffers and section 4.3 are unclear. ... I'm confused about this distinction. Does the offline replay buffer contain more than just the initial dataset?"
>
> Yes. The offline buffer contains online experiences and the initial offline dataset collected by other agents' behavior policies. The online buffer only stores the experiences collected by the agent's behavior policy.
>
> >"Why is it that REDQ fails completely on these tasks? ...there doesn't seem to be any improvement even in the online part for REDQ. Is the offline training a hindrance to the algorithm? "
>
> As we pre-train REDQ from the offline dataset, traditional online RL would suffer serious performance issues if the dataset is not diverse [REF1][REF2].
> However, when the dataset is diverse, REDQ would learn well. For example, in the case of a medium-replay dataset that consists of a relatively diverse dataset, REDQ can learn a good performance by offline pre-training (the starting performance is relatively high).
> When online interaction is available, REDQ or CQL would suffer serious extrapolation error [REF2] as the distribution of offline datasets may be very different from the online one.
> Figure 4 indicates that REDQ suffers more serious extrapolation error than CQL when an online interaction is available. CQL only suffers the serious stability issue on task halfcheetah-medium.
> That is why we need to maintain an online buffer to stabilize the current policy's learning.
>
> >"Would it perform better if it was simply initialized from scratch without the offline training? I think this would be a useful baseline to include too."
>
> We have conducted extra experiments on REDQ-online that learn from scratch without the offline training.
> To guarantee the reproduction performance, we use the official code published by the author instead of our implemented version.
> All hyperparameters follow the REDQ's default setting, e.g., the number of Qs is 10.
> The new results are included in Appendix F.
> From Figure 10, it can be observed that GCQL achieves a better or comparable performance than REDQ-online except for one task: the walker2d-random dataset.
> Specifically, when the dataset is good, e.g., including the expert dataset, GCQL can outperform the REDQ-online by a large margin.
> On the other hand,  even when the offline dataset is of poor quality, GCQL can still learn effectively by online interaction.
>
> >"CGQL-wo "without the online replay buffer where we only sample from the offline buffer"....The full GCQL would set W=0 when sampling from the offline buffer."
>
> Yes. All experiences get pooled into the offline buffer. We only remove the online buffer and reserve the conservative term.
> The full GCQL will set W=1 when sampling from the offline buffer to keep learning conservatively.
>
> >"Related to the previous point, how important is it that the online buffer is much smaller than the offline buffer?"
>
> The online buffer is for storing near-on-policy datasets. As [REF2] pointed out, extrapolation error is introduced by the mismatch between the dataset and true state-action visitation of the current policy.
> To keep the dataset's distribution in the online buffer close to the current policy's output distribution, we set it much smaller than the offline buffer to stabilize the learning process and speed up the convergence.
>
> >"To clarify, is the difference between REDQ and CGQL-wc (without the conservative term) only that the replay buffers are separated into two? Are all the other hyperparameters the same? Would this explain why REDQ fails but CGQL-wc can achieve decent performance?"
>
> The two-level buffer is the only main difference. Hence, GCQL-wc can benefit from the online buffer and alleviate the stability issue, while REDQ suffers from serious stability issue.
> In terms of the hyperparameters, the alpha in GCQL is updated following the CQL's method while alpha in REDQ is fixed, which follows the SAC's setting[REF4]. The other hyperparameters are the same.
>
> >"Do you have any idea what happened to CQL in halfcheetah-medium? In this environment, it seems like training more online actually leads to decreased performance over time, strangely enough."
>
> This is because the original CQL is trained on a stationary dataset that is fixed all the time. Now CQL is trained on a non-stationary dataset. Thus it may suffer from serious extrapolation error [REF2] as new different data are added to the replay buffer. This is why we need the online buffer to stabilize the learning process.
> The online buffer stores near-on-policy data, which is helpful to stabilize policy learning.

---

> > ### Author Response · Authors · 2021-11-18
> > **Responses to Reviewer VmYw - Continue**
> >
> > >"Looking at the ablations, it seems like the setting CGQL-wg does quite well overall. This would suggest that being more conservative the entire time is not a bad strategy. I'm a bit curious how these results compare to CQL."
> >
> > Even though GCQL-wg learns conservatively from both online and offline datasets, it still benefits from the online buffer as it can stabilize the learning process and speed up convergence [REF2].
> > Moreover, to make it clear, we conduct an extra ablation study on halfcheetah in appendix C.
> > We consider a new variant GCQL-wgo: GCQL without the greedy and online-offline two-level buffer, i.e., GCQL-wgo only samples from the offline buffer.
> > From Figure 7, it is clear that the greedy scheme and online buffer play an important role in boosting the performance and stabilizing the learning process when the dataset is sub-optimal.
> > For instance, when the online buffer is unavailable, both the GCQL-wgo and CQL suffer serious instability issues on the halfcheetah-medium task.
> > Such instability may be caused by the extrapolation error [REF2] introduced by the mismatch between the dataset and true state-action visitation of the current policy.
> >
> > >"In the ablations, CGQL-we does about as well as the full CGQL in every setting except walker2d-random. ... I'm not very convinced that adding ensembles is crucial here."
> >
> > Yes. The ensemble is not necessary for our framework.
> > As pointed out by reviewer qMWG, our framework can be easily adjusted to other offline methods, even policy penalty methods.
> > For instance, we have adjusted our greedy-conservative framework to TD3+BC [REF3] and boosted TD3+BC by a large margin for most tasks.
> > The new results are updated in Appendix D.
> >
> > >"Section 5.4. The results should be included in the main text, even if it is summarized. Currently, there are no results to look at. ... Also, I would add the label descriptions with the learning curves in the appendix for easier reading."
> >
> > Thank you for your valuable advice. We have updated our manuscript accordingly.
> > Due to the page number constraint, it is not easy to include all the figures in the main text.
> >
> > >"Moreover, how to evaluate the policy under the offline setting is also a challenging problem." While this is mentioned in the abstract..."
> >
> > The policy evaluation problem is naturally solved by online interaction as we can evaluate our policy by interacting with the environment.
> >
> > >"I'm not sure if "framework" is the proper word to use to describe adaptive Q-learning... Personally, I found this a bit confusing and had different expectations when reading the abstract."
> >
> > Our proposed adaptive Q-learning framework can be applied to other Q-learning-based methods, not just CQL. The general form is described by equation (2).
> > As we explained in section 4.4, the first term can be other value penalty terms, such as the divergence function between distributions over actions [REF5].
> > Moreover, suggested by reviewer qMWG, we can easily apply our framework to policy penalty methods, such as TD3+BC by changing equation (2) to $\pi^{k+1} \leftarrow \arg \max _{\pi^{k}} \mathcal{W}(s, a) \mathbb{B}\left(\pi^{k}\right)+\mathbb{A}\left(\pi^{k}\right)$,  in this case, the BC penalty term is the $\mathbb{B}\left(\pi^{k}\right)$ while $\mathbb{A}\left(\pi^{k}\right)$ is the TD3's policy normal objective.
> > We have conducted extra experiments to verify its efficiency in appendix D.
> >
> > >"Various typos and awkward sentences are scattered throughout and should be proofread."
> >
> > Thank you for the suggestion. We will update a proofread version.
> >
> > >"pre-existing offline data can prevent agents from converging prematurely due to the potential diverse, ...". Missing words?
> >
> > Due to the potential diverse offline dataset.
> >
> > [REF1] An Optimistic Perspective on Offline Reinforcement Learning.ICML.2020
> >
> > [REF2] Off-Policy Deep Reinforcement Learning without Exploration. ICML 2019
> >
> > [REF3] A Minimalist Approach to Offline Reinforcement Learning. arxiv.2021
> >
> > [REF4] Soft Actor-Critic: Off-Policy Maximum Entropy Deep Reinforcement Learning with a Stochastic Actor. ICML 2018.
> >
> > [REF5] Behavior regularized offline reinforcement learning.ArXiv,abs/1911.11361, 2019.

---

### Official Review · Reviewer_qMWG · 2021-11-03

**Correctness:** 4
**Technical Novelty And Significance:** 2
**Empirical Novelty And Significance:** 3
**Recommendation:** 6
**Confidence:** 3

**Main Review:**

I think the paper studies an important problem of offline-online RL where the algorithm has access to both offline data and a limited amount of online interactions. The authors empirically show the disadvantage of using a traditional approach of simply fine-tuning on the online data, as most offline RL algorithms like CQL will behave too conservatively. The approach, while simple, makes sense intuitively as it has the desired property of behaving pessimistically according to the offline data but greedily on the online data.

The proposed algorithm GCQL combines two specific offline RL algorithms CQL and REDQ. The central premise of the paper, to only act conservatively on the offline data, is a general concept that is agnostic to the specific training objective. I feel that the paper could be made more general by considering other forms of pessimism. For example, I would be interested in seeing if similar performance gains can be obtained by using a policy constraint penalty i.e. the penalty in TD3 + BC, rather than the value penalty in CQL. That way, the paper could be made more general by giving the practitioner the flexibility to use the best base learning algorithm for their specific problem domain.

One concern is that the performance gain of GCQL over GCQL-wg (which I interpret as effectively being CQL with ensemble Q-learning) appears to be very marginal. The only exception is when the offline dataset is random or highly suboptimal. In practice, it is common to assume that the dataset was generated by a decently-performing policy, so it is unclear if the novel training objective will yield much improvement. I think the paper would benefit from a more nuanced characterization by measuring the performance gap as a function of the suboptimality of the behavior policy.

**Summary Of The Paper:**

The paper proposes GCQL, a new RL algorithm that is trained on a mixture of offline data and online interactions. The novel component of the algorithm is a reweighting that balances acting pessimistically on offline data and greedily on online interactions. The authors propose a mixture replay buffer that consists of both offline and online samples. For online samples, the policy is trained as in the REDQ algorithm; meanwhile, for offline samples, the policy is additionally trained on a CQL-like value penalty. Finally, the authors show that GCQL outperforms existing SOTA offline RL algorithms that simply fine-tune on online data.


**Summary Of The Review:**

Overall, though the proposed algorithm seems like an incremental change over existing ones, it clearly is preferable to use over existing methods that simply fine-tune on the online data. I believe that the realization that using the same training objective for both offline and online data yields overly pessimistic policies is an important contribution, and opens the door for future work that considers different objectives for offline and online data as this paper does. Because of this, I recommend that the paper be accepted.

---

> ### Author Response · Authors · 2021-11-18
> **Responses to Reviewer qMWG**
>
> Thank you for your valuable comments. Below is our response to your concerns.
>
> > "...The central premise of the paper, to only act conservatively on the offline data, is a general concept that is agnostic to the specific training objective... i.e. the penalty in TD3 + BC, rather than the value penalty in CQL."
>
> Thanks for your constructive suggestion. We have conducted extra experiments on TD3+BC [REF1] following our greedy-conservative framework in appendix section D. The empirical results show that our greedy-conservative framework boosts the TD3+BC performance by a large margin for most tasks. We only test on the suboptimal dataset, i.e., random-v0, medium-v0, and medium-replay-v0, as these datasets are more challenging for offline RL methods.
> Moreover, applying our framework to the policy penalty method (TD3+BC ) is very easy where less than 20 lines codes are needed.
>
> >"One concern is that the performance gain of GCQL over GCQL-wg (which I interpret as effectively being CQL with ensemble Q-learning) appears to be very marginal..., so it is unclear if the novel training objective will yield much improvement..."
>
> Our framework consists of two key components: the greedy-conservative methods and the online-offline replay buffer. The GCQL-wg still benefits from the online-offline buffer.
> To make it clear, we conduct an extra ablation study on halfcheetah in appendix C.
> We consider a new variant GCQL-wgo: GCQL without the greedy and online-offline two-level buffer, i.e., GCQL-wgo only samples from the offline buffer.
> From Figure 7, it is clear that the greedy scheme and online buffer play an important role in boosting the performance and stabilizing the learning process when the dataset is suboptimal.
> For instance, when the online buffer is unavailable, both the GCQL-wgo and CQL suffer serious instability issues on the halfcheetah-medium task.
> Such instability may be caused by the extrapolation error [REF2] introduced by the mismatch between the dataset and true state-action visitation of the current policy.
>
> >"I think the paper would benefit from a more nuanced characterization by measuring the performance gap as a function of the suboptimality of the behavior policy."
>
> Thanks for your advice. We will work on it in our future work.
>
> [REF1].A Minimalist Approach to Offline Reinforcement Learning. arxiv.2021
>
> [REF2].Off-Policy Deep Reinforcement Learning without Exploration. ICML 2019.

---

### Official Review · Reviewer_tsK8 · 2021-11-03

**Correctness:** 3
**Technical Novelty And Significance:** 2
**Empirical Novelty And Significance:** 3
**Recommendation:** 6
**Confidence:** 4

**Main Review:**

There are two distinct contributions in this paper. The first is in the proposed training with experience replay approach, and the second is the propose algorithm which combines elements of CQL and REDQ. Unfortunately, I'm not sure the experimental results adequately explore either of these contributions. What the results show is their method doing better in some cases when using their training approach, but this approach itself is not validated by these results. Why not compare to other methods mentioned in the intro, such as Lee et al. 2021, using the training procedure they were designed for? The authors highlight differences between what they propose and the related work, but not all of those differences preclude comparing between the mentioned offline-online methods and the propose method. The results here don't allow us to determine whether this approach improves on existing approaches to offline-online RL.

As for the other main contribution, it's equally difficult to assess the value of this new CQL/REDQ hybrid because it is only compared using the proposed training method which might disadvantage the methods used as comparison. This wouldn't be an issue if the training method was shown as an important or superior approach for doing offline-online RL, but, as I argue above, I don't believe that the authors have provided any evidence for this. For instance, looking at REDQ's results in table 1, it seems like this training regime causes it to underperform based on the results the REDQ paper report given the same number of interactions (I could be mistaken given the results in this paper use a normalized score who definition is unknown to me).

In terms of writing, the paper contains many typos and incorrect word choice. I don't think this prevented me from understanding what was meant but this paper would benefit from some careful proofreading. Otherwise, the paper is well structured and mostly easy to follow.

My concerns about the experimental results make me doubt that this paper is ready for publication. The authors should make sure that their claim, that this training approach is a superior way of doing offline-online RL, is properly validated by the experimental results. It isn't sufficient to just propose a method that works better when using this training approach.

# Questions:

- Is my understanding correct that newly collected samples are added to both the online and offline buffers?

- Is the initial offline dataset not used to initialize the offline buffer?

- p. 7, “we report the max average score as performance”, this is ambiguous. As I see it, we have different points along the x-axis and different seeds. What are taking the maximum over? What are we averaging over?


# Comments:

- p. 4, it’s difficult to appreciate this illustrative example without some more context. How was the offline dataset generated? How are the offline and online datasets used, are they just concatenated? How many seeds were used and what are do the shaded areas represent? Why would it be fair to say that CQL failed if it was able to improve, albeit slowly? Why is this not due to the starting performance being already quite good? What kind of performance should we be expecting in this domain?

- p. 5, the term C should be defined here. I could not confidently find it in the cited work, and, regardless, it is central enough that is should be restated here to avoid any possible ambiguity.

- p. 5, “we randomly select two Q functions just following the REDQ’s setting”, this is unclear. Eq. (4) shows only a single action-value function or index. What is the meaning of the hat over this action-value function? One of the major contributions is proposed combination of parts of CQL and REDQ. I expect this terms to be explicitly defined here.

- p. 6, eq. (6), this isn’t a proper definition for a function. As I understand it, nothing prevents identical state-action tuples from being present in both buffers. The output of this function given that (s, a) is indeterminate. This function definition requires another argument.



# Minor comments and nitpicks:

- p. 1, "pre-existing offline data can prevent agents from converging prematurely due to the potential diverse”, the potential diverse? typo?

- p. 2, "the agent can achieve an expert policy using very few online interaction steps regardless of the quality of the offline dataset”, how can this possibly be true? Wouldn’t a degenerate dataset mean that all the learning needs to be done online, which would like require more than “very few” interactions?

- p. 3, later in the paper, policies are treated as probabilistic, e.g., eq. (3). The policy definition provided here is a deterministic mapping of states to actions.

- p. 3, "a corresponding value function”, this probably should be called the action-value function, not the value function as the term is usually used to refer the state-value function.

- p. 3, “off-policy RL algorithms maintain a growing replay buffer”, this is true only at the start of learning. Buffers are typically kept at a fixed size max size.

- p. 6, algorithm 1, the variable t is not defined and its value is never changed.

- p. 7, "The entire offline-gradient-update step is about 1 million, and the entire online exploration step is around 90K”, 1 million what? What are we counting, what are the units?

- p. 7, table 1, what do the +/- ranges represent?

Post rebuttal
==============

The authors have improved the empirical evaluation and have mostly addressed my biggest concerns. I have increased my scores to reflect this.


**Summary Of The Paper:**

The authors propose a mixed offline-online RL approach for which they design an algorithm. They propose to maintain 2 separate replay buffers, one for online data and one for offline data, to allow them to sample either an online or offline batch of data when doing an update step, and tailor the loss function based on the batch's provenance. In addition, the authors propose using a CQL variant which uses an ensemble of action-value function as done by REDQ to learn in this setting. They conclude by showing some empirical results on the D4RL Mujoco benchmark domains.

**Summary Of The Review:**

The authors shows that the proposed method performs better under their proposed training regime but don't provide any supporting evidence for this training regime. This raises some concerns about the significance of the empirical results, and the usefulness of the proposed regime. These concerns prevent me from recommending acceptance.

---

> ### Author Response · Authors · 2021-11-18
> **Responses to Reviewer tsK8**
>
> Thank you for your valuable comments. Below is our response to your concerns.
>
> > "...Why not compare to other methods mentioned in the intro, such as Lee et al. 2021, using the training procedure they were designed for?"
>
> Lee et al.[REF1] did not provide the pre-training code for the ensemble CQL critics. Their CQL critic neural network architecture is customized, different from the original ones, e.g., changing the hidden layer's number from 2 to 3.
> Hence, we directly use their pre-trained CQL critics and the fine-tuning code provided by the author to guarantee the reproduction performance.
> In this experiment, we followed their default setting except for the total online exploration steps.
> The learning curves are updated in the appendix section E.
> The results indicated that our method still achieved a comparable or better performance than theirs.
>
> Moreover, our proposed method is much simpler and robust to the hyperparameters shown in section 5.4.
> In contrast, Lee's method used a customized network architecture, and the hyperparameters are fine-tuned.
>
> Furthermore,  our method targets a foundational problem: effectively utilizing the offline and online datasets as they have different advantages. Therefore, we should not simply stress one and ignore another.
> In contrast, Lee et al. emphasize the online data and utilize the offline dataset for selecting near-on-policy samples to alleviate the distribution shift issue [REF1].
>
> Thirdly, our framework is very general and can be easy adjusted to other methods.
> For instance, as reviewer qMWG pointed out, we apply our framework to a policy penalty method: TD3+BC[REF2], by modifying their policy objective from
> $\pi=\text{argmax}\_{\pi} \mathbb{E}\_{(s, a) \sim \mathcal{D}}\left[\lambda Q(s, \pi(s))-(\pi(s)-a)^{2}\right]$ to $\pi=\text{argmax}\_{\pi} \mathbb{E}\_{(s, a) \sim \mathcal{D}}\left[\lambda Q(s, \pi(s))-\mathcal{W}(s,a)(\pi(s)-a)^{2}\right]$.
> Our extra experiments on TD3+BC in appendix section D indicated that our framework can boost TD3+BC's performance by a large margin when an online interaction is available.
>
> > "The authors highlight differences between what they propose and the related work, ...The results here don't allow us to determine whether this approach improves on existing approaches to offline-online RL."
>
> As explained above, we have conducted extra experiments to compare Lee's [REF1] methods. We have updated our paper to include these extra experiments and results in Appendix E. These results indicate our methods still achieved a comparable or better performance than theirs.
> Moreover, the AWAC[REF3] is another offline-online RL baseline whose score has been included in table 1.
> AWAC employs 500K online exploration steps while ours only employ 90K and gain better performance than AWAC.
>
> >"..(I could be mistaken given the results in this paper use a normalized score who definition is unknown to me)."
>
> The normalized score follows the definition of D4RL[REF4], where a score of 100 corresponds to the average returns of a domain-specific expert. In contrast, a normalized score of 0 corresponds to the average returns (over 100 episodes) of an agent taking actions uniformly at random across the action space. We have added an explanation on it in the caption of Table 1.
>
> > "Is my understanding correct that newly collected samples are added to both the online and offline buffers?"
>
> Yes. As we explained in section 4.3, the online buffer store near-on-policy data for stabilizing the learning process. That is why we need to keep it relatively small size. By contrast, the offline buffers store all samples.
>
> > "Is the initial offline dataset not used to initialize the offline buffer?"
>
> As we explained in section 4.3, the offline buffer stores all experiences, including samples collected by its behavior policy or any other agent's behavior policy (i.e., the offline dataset). Hence, the initial offline dataset is used to initialize the offline buffer.
>
> > "we report the max average score as performance", this is ambiguous. As I see it, we have different points along the x-axis and different seeds. What are taking the maximum over? What are we averaging over?"
>
> The maximum is the max score for each seed. and we average across different seeds.
> We have updated it accordingly in section 5.2.
>
>  > "... How was the offline dataset generated?"
>
>  The offline dataset comes from the commonly used benchmark D4RL[REF4] for offline RL.
>
> > "...How are the offline and online datasets used, are they just concatenated?"
>
> As explained in section 4.3, we employ a two-level replay buffer: online buffer and offline buffer. The online buffer stores near-on-policy data collected by interaction with the environment, while the offline buffer stores all data, including the offline dataset provided by the D4RL [REF4] benchmark. When updating the policy, we randomly sample from these two buffers. In our experiments, we sample from the online buffer following a possibility of 0.5.

---

> > ### Author Response · Authors · 2021-11-18
> > **Responses to Reviewer tsK8 - Continue**
> >
> > > "How many seeds were used and what are do the shaded areas represent? "
> >
> > As indicated by the caption of Figure 4, we used five random seeds, and the shared areas represent the standard deviation across different seeds.
> >
> > > "Why would it be fair to say that CQL failed if it was able to improve, albeit slowly?"
> >
> > As indicated in section 4.1, we expect offline RL or online RL to learn effectively from the fresh online data when offline data is suboptimal. However, the CQL failed to learn effectively from the online data as the learning curves' improvements of CQL are very limited when online exploration is available.
> >
> > > "Why is this not due to the starting performance being already quite good? What kind of performance should we be expecting in this domain?"
> >
> > In most tasks, the starting performance is not quite good (worse than medium performance, i.e., 50). We expect the agent to yield a performance that is close to or better than the expert policy, whose normalized score is 100.
> >
> > > "the term C should be defined here. I could not confidently find it in the cited work, and, regardless, it is central enough that is should be restated here to avoid any possible ambiguity."
> >
> > The conservative term used in CQL($\mathcal{H}$) (the variant used in our implementation) can be presented by
> > $$
> > \alpha \mathbb{E}\_{\mathbf{s} \sim \mathcal{D}}\left[\log \sum\_{\mathbf{a}} \exp (Q(\mathbf{s}, \mathbf{a}))-\mathbb{E}\_{\mathbf{a} \sim \hat{\pi}\_{\beta}(\mathbf{a} \mid \mathbf{s})}[Q(\mathbf{s}, \mathbf{a})]\right]
> > $$
> > In our case, the first $\mathbf{a}$ used in the left Q funtion is sampled from current policy, i.e., $\mathbf{a} \sim \pi\_{\phi}(\cdot|s),s\sim\mathcal{D}\_{\text{OORB}}$, while the $\mathbf{a}$ used in the right Q function is sampled from the dataset, i.e., $\mathbf{s},\mathbf{a} \sim \mathcal{D}\_{\text{OORB}}$. Here, $\hat{\pi}\_\beta$ means the behaviour policy. We updated our manuscript accordingly.
> >
> > > "we randomly select two Q functions just following the REDQ's setting", this is unclear. Eq. (4) shows only a single action-value function or index. What is the meaning of the hat over this action-value function?"
> >
> > We randomly select two Q functions from the Q function ensemble, and the $\mathcal{M}$ represents the index of the randomly selected Q functions.
> > Here, the hat means the target Q function, as Q-learning-based methods usually employ two Q functions to stabilize the learning process [REF5].
> > We have added more details on this part in section 4.4.
> >
> > >"One of the major contributions is proposed combination of parts of CQL and REDQ. I expect this terms to be explicitly defined here."
> >
> > The combination is just one implementation that followed our framework and it is very easy to apply our framework to other value penalty methods [REF10] or even the policy penalty method. As mentioned above, we adjusted our framework to TD3+BC [REF2] without any fine-tuning and gained a clear performance improvement when an online interaction is available. The experiment details are presented in Appendix D.
> >
> > >"p. 6, eq. (6), this isn’t a proper definition for a function. As I understand it, nothing prevents identical state-action tuples from being present in both buffers. The output of this function given that (s, a) is indeterminate"
> >
> > Sorry for the confusion. We have updated its definition accordingly.
> >
> > > "pre-existing offline data can prevent agents from converging prematurely due to the potential diverse", the potential diverse? typo?"
> >
> > It is a typo. It should be diverse offline dataset.
> >
> > > "p.2, "the agent can achieve an expert policy using very few online interaction steps regardless of the quality of the offline dataset", how can this possibly be true?
> >
> > Sorry for the confusion. We mean that we can use much less online interaction but achieve a similar or even better performance than previous SOTA online methods.  For instance, shown in Appendix F, we outperform the SOTA online method: REDQ [REF9] by a large margin for most tasks. We have revised this statement in the new version of our manuscript.

---

> > > ### Author Response · Authors · 2021-11-18
> > > **Responses to Reviewer tsK8 - Continue**
> > >
> > > > "Wouldn't a degenerate dataset mean that all the learning needs to be done online, which would like require more than "very few" interactions?"
> > >
> > > As discussed in section 1, the degenerate dataset can improve the samples' potential diversity, which is important for RL learning [REF8].
> > > For example, the halfcheetah-random-v0 consists of data collected by a random policy.
> > > However, such a "random" dataset can still enable our method to achieve an expert policy within 100K online interactions.
> > > In contrast, conventional online RL methods, such as TD3 [REF7] or SAC [REF6], need at least 1 million online interaction steps to gain an expert policy.
> > > Hence, we set the sample possibility from the online buffer to 0.5 for all tasks in our experiments.
> > > In another word, we take the near-on-policy data equally important with the offline dataset.
> > > On the other hand, as we showed in the hyperparameters analysis section 5.4, more offline updates perform better when dataset quality is very good or bad.
> > > On the contrary, more online updates are preferred for the medium-quality dataset.
> > > In sum, setting sample possibility to 0.5 may be not the best option for every task but works well in general in our experiments.
> > >
> > >  > "p.3, later in the paper, policies are treated as probabilistic, e.g., eq. (3). The policy definition provided here is a deterministic mapping of states to actions."
> > >
> > >  We do not restrict the policy to be deterministic in our work. We have added some explanations to clarify it in Sec. 3.
> > >
> > > > "p. 3, "a corresponding value function”, this probably should be called the action-value function, not the value function as the term is usually used to refer the state-value function."
> > >
> > > Thanks for pointing it out. We have updated it accordingly in the new version of our manuscript.
> > >
> > > >"p. 3, “off-policy RL algorithms maintain a growing replay buffer”, this is true only at the start of learning. Buffers are typically kept at a fixed size max size."
> > >
> > >  Sorry for this unclear statement. We have modified it to "maintain a replay buffer" in the revisited version.
> > >
> > > >"p. 6, algorithm 1, the variable t is not defined and its value is never changed."
> > >
> > > Here, t counts the offline update steps. We have clearly defined it in the latest version.
> > >
> > > >"p. 7, "The entire offline-gradient-update step is about 1 million, and the entire online exploration step is around 90K”, 1 million what? What are we counting, what are the units?"
> > >
> > > It is 1 million offline update steps, i.e., 1 million is the total number of offline gradient update steps. We have updated it accordingly in the revisited version.
> > >
> > >  >"p. 7, table 1, what do the +/- ranges represent?"
> > >
> > >  +/- captures the standard deviation over seeds. We have revised the paper accordingly.
> > >
> > >  [REF1]  Offline-to-Online Reinforcement Learning via Balanced Replay and Pessimistic Q-Ensemble. CoRL 2021.
> > >
> > >  [REF2]  A Minimalist Approach to Offline Reinforcement Learning. arxiv.2021.
> > >
> > >  [REF3]. Accelerating Online Reinforcement Learning with Offline Datasets. CoRR,2020.
> > >
> > >  [REF4]. D4rl: Datasets for deep data-driven reinforcement learning,arxiv,2020.
> > >
> > >  [REF5]. Human-level control through deep reinforcement learning. Nature. 2015.
> > >
> > >  [REF6]. Soft actor-critic: Off-policy maximum entropy deep reinforcement learning with a stochastic actor. ICML.2018
> > >
> > >  [REF7]. Addressing Function Approximation Error in Actor-Critic Methods. ICML.2018.
> > >
> > >  [REF8]. An optimistic perspective on offline reinforcement learning. In ICML, 2020.
> > >
> > >  [REF9]  Randomized Ensembled Double Q-Learning: Learning Fast Without a Model. ICLR,2021.
> > >
> > > [REF10] Behavior regularized offline reinforcement learning.ArXiv,abs/1911.11361, 2019.

---

> > ### Comment · Reviewer_tsK8 · 2021-11-19
> > **Reply to authors**
> >
> > I thank the authors for the detailed response and numerous clarifications. I will be increasing my score and editing my review to reflect this. I will put all my comments for their response here to avoid unnecessary indentation due to nested replies.
> >
> > > As we explained in section 4.3, the offline buffer stores all experiences, including samples collected by its behavior policy or any other agent's behavior policy (i.e., the offline dataset). Hence, the initial offline dataset is used to initialize the offline buffer.
> >
> > I see. I missed or forgot that sentence during my read-through. I think Alg. 1 could be made more informative, e.g., have variables for the buffers and show explicitly some of the initializations like $D_{offline} \leftarrow D_{initial}$. My confusion came from trying to find this information in the pseudocode.
> >
> > > The maximum is the max score for each seed. and we average across different seeds. We have updated it accordingly in section 5.2.
> >
> > Am I correct to assume that this only applied to table 1? If so, I would say this explicitly since I was under the impression that it applied to all reported results which was the source of my confusion. If not, then it is still unclear to me.
> >
> > > As indicated by the caption of Figure 4, we used five random seeds, and the shared areas represent the standard deviation across different seeds.
> >
> > Except I was referring to Figure 2. This might be generated by the same data but that is not clear to the reader. The reader shouldn't be expected to look at the caption of Figure 4 for information about the Figure 2.
> >
> > > In most tasks, the starting performance is not quite good (worse than medium performance, i.e., 50).
> >
> > My comments here were specific to the "illustrative example". If think the authors have misinterpreted this particular comment. My concern is that the example is difficult to appreciate due to limited context, like understanding that 100 is achievable and that 50 is considered poor. I would encourage the authors to somewhat rework the presentation of Sec. 4.1 so that it is easier to appreciate the point the authors are trying to make.
> >
> > > In our case, the first $\mathbf{a}$ used in the left Q funtion is sampled from current policy [...]
> >
> > The scope of the $\sum_\mathbf{a}$ is ambiguous. I would add some parenthesis/bracket to avoid any confusion about the meaning of the second $\mathbf{a}$.

---

> > > ### Author Response · Authors · 2021-11-20
> > > **Responses to Reviewer tsK8**
> > >
> > > Thank you for your valuable comments. Below is our response to your concerns.
> > >
> > > >... e.g., have variables for the buffers and show explicitly some of the initializations like $D_{offline}\gets D_{initial}$....
> > >
> > > Thank you for your valuable advice. We have added more explanations about buffers in Alg. 1.
> > >
> > > >Am I correct to assume that this only applied to table 1? If so, I would say this explicitly since ...
> > >
> > > Yes, this is only applied to Table 1. We take maximization over testing scores during the whole learning process and average these max scores coming from different random seeds. Other results in the paper are shown via learning curves in figures, and thus, only averaging is needed and applied for them. Sorry for the confusion. We have added more explanations to clarify this in Sec. 5.2 in our updated version.
> > >
> > > >Except I was referring to Figure 2. ...
> > >
> > > Thanks for pointing it out. We have added more explanation at the caption of Figure 2.
> > >
> > > >...I would encourage the authors to somewhat rework the presentation of Sec. 4.1 so that it is easier to appreciate the point the authors are trying to make.
> > >
> > > Sorry for our misunderstanding. We have reworked Sec 4.1 to address the concern.
> > >
> > > >The scope of the $\sum_a$is ambiguous. I would add some parenthesis/bracket to avoid any confusion about the meaning of the second $a$.
> > >
> > > Thank you for your advice. We have used another notation $\dot{a}$ in Eq. (3) in the paper to resolve this ambiguity. The new  $\sum_a$ is presented by
> > > $
> > > \sum\_{\mathbf{\dot{a}}} \exp (Q(\mathbf{s}, \mathbf{\dot{a}}))
> > > $

---

### Official Review · Reviewer_irZQ · 2021-11-06

**Correctness:** 3
**Technical Novelty And Significance:** 2
**Empirical Novelty And Significance:** 2
**Recommendation:** 3
**Confidence:** 4

**Main Review:**

Poor writing. Inadequate experiments design. This paper, at its current status, is not ready to be submitted to conferences like ICLR.
Actionable feedback for the authors might be rethinking the storyline of this work.

**Summary Of The Paper:**

This paper presents a framework called adaptive Q-learning that integrates the advantage of offline learning and online learning.

**Summary Of The Review:**

see above.

---

> ### Author Response · Authors · 2021-11-17
> **Responses to Reviewer irZQ**
>
> Thank you for your valuable comments. Below is our response to your concerns.
> > "...Poor writing."
>
> We have uploaded a new version of this paper with more careful proofreading. Besides, we would like to re-emphasize our contributions here.
> - To address the issues existing in offline and online RL, we propose a unified and simple framework that can effectively benefit from both the offline dataset and limited online data and gain a high sample efficiency.
> - To implement the above framework, we propose a practical algorithm called Greedy-conservative Q-ensemble learning (GCQL) based on SOTA offline and online RL methods.
> We also show that our framework can be easily implemented with other RL algorithms (see Appendix D in the paper).
> - We empirically verify the effectiveness of our algorithm by comprehensive experiments on the widely used locomotion benchmark MuJoCo [REF1].
>
> We hope this can resolve your concerns.
>
> > "...Inadequate experiments design."
>
> We are not sure if we understand the reviewer's concern correctly.
> We are open to further clarifying if the reviewer could provide concrete questions on the experiment design.
>
> Moreover, we would like to recapitulate our experiment design and results here.
>
> First, we choose popular offline benchmark D4RL [REF2] as the offline dataset and MuJoCo [REF1] simulator as the online environment. These tasks are recognized and used by many previous works.
>
> Second, we select various SOTA baselines, including offline RL method CQL [REF5], online RL method REDQ[REF6], offline-online methods AWAC [REF3], and off2on [REF4]. Superior experimental performances demonstrate that our algorithm can effectively make use of both offline and online data and improve the sample efficiency of current methods.
>
> Finally, we conduct ablation studies to check the importance of each component in our framework in Sec. 5.3, and we also present an analysis for hyperparameters to show the robustness of our algorithm in Sec. 5.4.
>
> [REF1] Mujoco: A physics engine for model-based control. IEEE. 2012.
>
> [REF2] D4rl: Datasets for deep data-driven reinforcement learning,arxiv,2020.
>
> [REF3] Accelerating Online Reinforcement Learning with Offline Datasets. CoRR,2020.
>
> [REF4] Offline-to-Online Reinforcement Learning via Balanced Replay and Pessimistic Q-Ensemble. CoRL 2021.
>
> [REF5] Conservative Q-Learning for Offline Reinforcement Learning. NeurIPS2020.
>
> [REF6] Randomized Ensembled Double Q-Learning: Learning Fast Without a Model. ICLR,2021.

---

### Author Response · Authors · 2021-11-18
**Revision Summary**

We thank all reviewers for the careful and constructive reviews. We revised the paper accordingly and marked the major modifications in blue for visibility. The major changes are summarized as follows.
- We have added some ablation studies on halfcheetah to more deeply investigate our proposed algorithm in Appendix C (qMWG&VmYw).
- We have added the experiments about adjusting our framework to the policy penalty method (TD3+BC) in Appendix D (qMWG).
- We have updated our main experiment results by adding a baseline off2on which also focuses on offline-online RL in Appendix E (tsK8).
- We have added experiment results for the online RL algorithm REDQ in Appendix F (VmYw).
- We have fixed the typos and addressed other comments mentioned by the reviewers.

We look forward to further discussions and feedbacks.

---

### Decision · Program_Chairs · 2022-01-20

**Decision:**

Reject

**Comment:**

In this paper, the authors studied reinforcement learning applications that have access to both online and offline data (with limited online interaction though).  In order to handle the mixture of online and offline data efficiently, the authors proposed a new paradigm called adaptive Q-learning, which treats offline and online data differently (as reflected by whether pessimism is implemented or not). The effectiveness of the proposed paradigm has been tested empirically. The reviewers have raised concerns about the sufficiency and significance of the experiments conducted in the paper, and pointed out that the proposed algorithmic idea is a somewhat incremental change over existing ones. The changes the authors promised to make will make the paper stronger.